# Measuring and Validating the Factors Influenced the SME Business Growth in Germany—Descriptive Analysis and Construct Validation

**Hosam Azat Elsaman** [1,*] , **Nourhan El-Bayaa** [2] **and Suriyakumaran Kousihan** [1]

1 York Business School, York Saint John University, York YO31 7EX, UK
2 Edinburgh Business School, Heriot-Watt University, Edinburgh EH14 4AS, UK
* Correspondence: hosam.elsaman@yorksj.ac.uk

**Abstract:** In Germany, the medical device industry constitutes a cornerstone of the health sector. In this study, we investigated the challenges and factors affecting the present-day performance of German SMEs concerned with medical devices. The research methodology adopted a cross-sectional and correlational research design, with simple random-sampling techniques, to data obtained from 110 mid-level and senior managers in German SMEs by means of an online structured survey in August 2022. We statistically validated our study data using exploratory factor analysis (EFA), Kaiser–Meyer–Olkin (KMO) testing, and Bartlett's test, to assess the relationship between study variables and measure data adequacy using the R4.1.1(21) software, then carried out principal component analysis (PCA) with varimax factor loading and extracted six factors for use as research variables. The researchers also applied descriptive data analysis techniques using SPSS.21. The main study variables were: (1) the business performance of small and medium businesses (SMP); (2) their financial situation (SMEF); and (3) their implementation of new medical device industry regulations (MDR). By such statistical means, results confirmed poorer business performance and lower anticipated growth amongst SMEs affected by MDR, over and above the impacts of the present-day economic situation. The data can be used by management information systems (MIS) and decision system support professionals for planning and developing practical models about how to cope with current industry challenges. We recommend further research involving inferential analysis and triangulation of these data in the form of a semi-structured qualitative study in the larger scope of the population and different sectors.

**Dataset:** Please see the supplementary file.

**Dataset License:** CC-BY

**Keywords:** medical devices; challenges; descriptive statistics; exploratory factor analysis

## 1. Summary

The EU regulatory system conducted modification in the medical device industry by implementing new regulation policies and restrictions with a very short-term deadline and limited resources to facilitate the transformation for companies especially SME in Europe. In this study, we collected primary data and carried out robust empirical statistical validation tests to identify relationships among study variables. We sought to highlight the impact of new medical device reporting (MDR) regulation on the business performance and strategy of SMEs in the southern German regions of Bavaria and Baden-Württemberg in August 2022, one year after the new regulations were introduced. These two regions were chosen because most SMEs working in the healthcare and medical devices sector are located in these areas. We identified a total of 467 medical device companies. For our research, we formulated a Likert Scale design survey including three sets of variables with a total

of 29 factors, The first part of the questionnaire was concerned with SME implementation of the new regulations. The second part of the questionnaire was designed to assess the organizational performance of SMEs within the context of corporate development in light of existing challenges. We sought to measure mentoring areas and the consequences of applying the new restrictions in terms of personnel organization, supervision, and development, as well as effects on scientific research [1]. The third part of our survey related to the financial difficulties and marketing burdens faced by SMEs. We carried out instrument validation by means of exploratory factor analysis (EFA), Kaiser–Meyer–Olkin (KMO) testing, and Bartlett's test, to determine relationships among study variables and measure data adequacy [2]. We carried out principal component analysis (PCA) with varimax factor loading and extracted six factors for use as research variables. We then applied descriptive data analysis. By such statistical means, we identified poorer business performance and lower anticipated growth amongst SMEs affected by MDR, over and above the impacts of the present-day economic situation.

This study is the first empirical assessment of the implications of the new MDR (2017/745) regulations on SMEs involved with medical devices in the southern German regions of Bavaria and Baden–Württemberg, one year after the regulations entered into force. We collected primary data and carried out robust empirical statistical validation tests to identify relationships among study variables. Our findings may enable SMEs to reconsider their future business strategies, and also prompt further research involving inferential analysis and triangulation of these data in the form of a semi-structured qualitative study [3].

## 2. Data Description

In this study of SMEs, we computed data for the three variables of business performance (SMP), financial situation (SMF), and implementation of new medical regulations (MDR); first, by EFA, and then by descriptive statistical techniques, as follows.

### 2.1. Polychoric Correlation Matrix

Polychoric correlation measures the link between instrument variables. In this study, we determined a value of 0.9 as shown in Figure 1, which clearly indicated the validity of research instrument variables for factor analysis computation purposes [4].

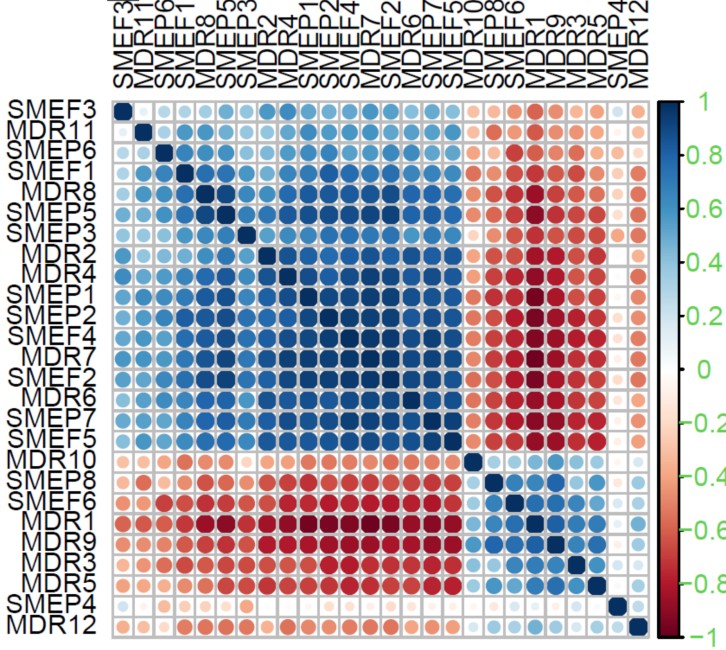

**Figure 1.** Polychoric correlation for variables computed by the author using R software 4.1.1 (21).

2.1.1. Kaiser Measure of Overall (KMO) Sampling Adequacy

Bartlett's test results should be validated with a measure of sampling adequacy because the sample adequacy of larger data sets is highly sensitive to even minor deviations. The Kaiser–Meyer–Olkin (KMO) test is used to determine the appropriateness of data for factor analysis. It measures the extent to which the indicators of the construct belong together. Ideally, the KMO measure should be greater than 0.80; however, a figure above 0.60 is tolerable. The overall KMO measure can sometimes be increased by deleting the offending variables whose KMO value is low. Because each set of variables in a relatively homogenous set measures the same basic concepts or categories, high correlations between variables demonstrate that the variables can be homogeneously categorized. Bartlett's test of sphericity can be used to test the factorability of the correlation matrix, which can then be evaluated by statistical means [5].

The result of the KMO test in Table 1 was 0.883, which reflected strong significance since it was higher than the significance value of 0.80. For Bartlett's test, the result detected $p = 0.000$, a strong significance as it was below the significance value of 0.05. These values refer to the adequacy of the sample for the KMO test, and to the applicable-factors correlation matrix in the case of Bartlett's test.

**Table 1.** KMO and Bartlett's test results.

| Kaiser–Meyer–Olkin Measure of Sampling Adequacy | | 0.883 |
|---|---|---|
| Bartlett's test of sphericity | Approx. chi-square | 144.905 |
| | df | 21 |
| | Sig. | 0.000 |

2.1.2. Principal Component Analysis

We used principal component analysis (PCA) to detect the existence of latent variables in our instrument data. These latent variables are often referred to as factors and dimensions. When applying the method of varimax rotation, the major objective is to produce a factor structure in which each variable loads. Such a factor structure will result in each factor representing a distinct construct. Factor analysis of our study variables produced six factors—two for each variable—as shown in Tables 2 and 3. For the financial situation of our study (SMEF), factor loading extracted (1) financial performance and (2) commercial performance. For business performance (SMP), factor loading extracted (1) innovation strategies and (2) business growth, as shown in Tables 4 and 5. Concerning business responses to (MDR), factor loading extracted the factors of (1) MDR implementations and (2) transparency, as shown in Tables 6 and 7.

**Table 2.** Factor loading for financial situation SMEF results.

| Variable | Components | |
|---|---|---|
| | Financial Performance | Commercial Performance |
| SMEF4 | 0.890 | −0.109 |
| SMEF2 | 0.843 | −0.169 |
| SMEF1 | 0.619 | |
| SMEF5 | 0.612 | |
| SMEF3 | 0.417 | |
| SMEF6 | −0.112 | 0.993 |

**Table 3.** Variance and rotation sums of squared loadings for financial performance and commercial performance results.

| Variables | Total | % of Variance | Cumulative % |
|---|---|---|---|
| Financial performance | 1.684 | 41.661 | 41.661 |
| Commercial performance | 0.914 | 22.625 | 64.285 |

**Table 4.** Factor loading for MDR implementation results.

| Variables | Components | |
| :---: | :---: | :---: |
| | **MDR Implementation** | **Transparency** |
| MDR7 | 0.804 | −0.371 |
| MDR1 | −0.690 | 0.345 |
| MDR11 | 0.678 | 0.252 |
| MDR6 | 0.666 | −0.262 |
| MDR4 | 0.655 | −0.162 |
| MDR8 | 0.596 | |
| MDR2 | 0.524 | −0.311 |
| MDR9 | | 0.918 |
| MDR5 | | 0.567 |
| MDR3 | −0.208 | 0.384 |

**Table 5.** Variance and rotation sums of squared loadings for MDR implementation results.

| Variables | Total | % of Variance | Cumulative % |
| :---: | :---: | :---: | :---: |
| MDR Implementations | 3.579 | 35.786 | 36.184 |
| Transparency | 1.383 | 13.834 | 53.975 |

**Table 6.** Factor loading for SMP results.

| Variables | Components | |
| :---: | :---: | :---: |
| | **Innovation** | **Business Growth** |
| SMEP2 | 0.763 | −0.124 |
| SMEP1 | 0.717 | −0.264 |
| SMEP5 | 0.698 | |
| SMEP3 | 0.657 | |
| SMEP7 | 0.649 | −0.226 |
| SMEP6 | 0.461 | 0.135 |

**Table 7.** Variance and rotation sums of squared loadings for SMP results.

| Variable | Total | % of Variance | Cumulative % |
| :---: | :---: | :---: | :---: |
| Innovation | 2.265 | 35.826 | 35.826 |
| Business growth | 1.529 | 24.183 | 60.009 |

2.1.3. Descriptive Analysis Findings

For each part of the study survey, data were collected, classified, and coded. We statistically computed the dataset using the Statistical Package for Social Science (SPSS) version 21 and R software 4.1.1. The research study design involved descriptive and inferential statistical methods. The descriptive analysis included frequency distribution and percentage calculations, as well as the measurement of central tendency by calculating mean, standard deviation, skewness, and kurtosis values for the collected dataset. The inferential analysis involved the determination of the multicollinearity regression for the independent variable.

As shown in Table 8, the most frequently reported years of experience among respondents was 11–15 years, with a percentage of 48% and frequency N = 47, followed by 16–25 years, with a percentage of 20%, and frequency N = 22. In terms of respondents' business responsibilities, regulatory affairs were most common, with a percentage of 38%, and frequency of N = 43; followed by management, with a percentage of 22% and frequency of N = 25; followed by sales, production, and business owners, with percentages of 17%, 13%, and 10%, respectively.

**Table 8.** Descriptive statistics (frequency and percentage) of professional characteristics of respondents.

| Professional Characteristics of Respondents | Frequency | Percent |
|:---:|:---:|:---:|
| Years of experience | | |
| 1–5 years | 13 | 12% |
| 6–10 years | 20 | 18% |
| 11–15 years | 47 | 42% |
| 16–25 years | 22 | 20% |
| 25 years plus | 10 | 8% |
| Department in company | | |
| Sales | 19 | 17% |
| Production | 14 | 13% |
| Management | 25 | 22% |
| Regulatory affairs | 43 | 38% |
| Business owner | 11 | 10% |

Table 9 shows that, for financial performance variables, agreeableness has a mean score (1.85). This means that employees of SMEs agreed to the negative financial impact of applying MDR by taking on extra burdens. An SD of 0.757 indicates a low level of data distribution and good reliability. SD values above 1 indicate higher data spread and lower reliability, but this was not the case with our data [5]. However, our skewness and kurtosis values were both above 1. This result shows that our data were not normally distributed, and that our dataset was slightly skewed. The skewness value of 1.484 indicated a right-skewed data distribution, and the kurtosis value of 4.664 indicated a distribution of data that was heavily tailed in comparison with normal distribution.

**Table 9.** Central tendencies of descriptive statistics.

| Variables | Mean | Standard Deviation (SD) | Skewness | Kurtosis |
|:---:|:---:|:---:|:---:|:---:|
| Financial performance | 1.85 | 0.757 | 1.484 | 4.664 |
| Commercial performance | 2.11 | 0.870 | 1.117 | 1.680 |
| Innovation | 2.05 | 0.656 | 0.993 | 3.672 |
| Business growth | 1.70 | 0.919 | 1.610 | 2.691 |
| Transparency | 2.48 | 0.878 | 0.378 | 1.97 |
| MDR Implementation | 4.07 | 0.869 | −1.565 | 3.500 |

In terms of commercial performance, a mean value of 2.11 suggested that the major effect of the new MDR regulations on SMEs was reduced market share and sales capacity in Germany and Europe. An SD value of less than 1 (0.870) indicated that the data were closely clustered around a more reliable mean. while skewness and kurtosis values above 1, of 1.117, and 1.680, respectively, indicated that the data were not normally distributed.

Among the descriptive results for the innovation factor, a mean value of 2.05 confirmed the feedback we obtained from our respondents that the new MDR did not affect innovation or creativity in the medical devices sector in southern Germany. In addition, SD, skewness, and kurtosis values, of 0.656, 1.117, and 1.680, respectively, indicated a close spread of data around the mean and a normal data distribution.

For the variable of business growth, a mean value of 1.70 supported the solid consensus of our respondents that the implementation of the new MDR regulation produced a negative effect on the business growth strategies of SMEs. Indeed, several companies reported plans to exit the medical devices sector as a result of the new restrictions. SD, skewness, and kurtosis values of 0.919, 1.610, and 2.691, respectively, indicated a close spread of data around mean, and normal data distribution patterns in terms of skewness and kurtosis.

The findings for the independent variable of MDR implementation highlighted the strongly negative feedback we obtained from our sample respondents concerning the implementation of new MDR with respect to such matters as the number of regulatory bodies involved. A mean value of 4.07 strongly confirmed this. Values for SD, skewness, and kurtosis of 0.869, 1.565, and 3.50 indicated normal data distribution for skewness and

kurtosis, The skewness value that indicated the data distribution was right-skewed, and the kurtosis value indicated a heavily tailed data distribution, in comparison with normal distribution. The SD value indicated a clustered distribution of data around the mean.

Conversely, for the transparency factor, we obtained a mean value of 2.48, which was in line with the broadly neutral feedback which we received from our respondents about the effects of the new MDR on transparency procedures in the medical devices industry. This matter might be more thoroughly investigated by extra inferential analysis between research variables. The skewness proportion of the major dataset is slightly skewed which means it is suitable for parametric data testing by means of Pearson correlation because the number of responses in the sample is higher than 100. A Spearman correlation might also be applied in any inferential statistical analysis.

### 2.1.4. Multicollinearity Test

Multicollinearity variables can be found in a regression model if the independent variables are correlated with each other; in such a case, the dependent variables' relation is very predictable and can lead to inaccurate statistical results [6]. For this reason, in our study, before conducting regression and correlation analysis, we computed the multicollinearity regression for the independent variables [7]. We found variances in inflation factor (VIF) values of approximately one for both independent variables, as shown in Table 10. This result indicated the absence of variables correlation, and so our variables were suitable for computing correlation and regression tests.

**Table 10.** Multicollinearity coefficient test.

| | | Coefficients [a] | |
|---|---|---|---|
| | | **Collinearity Statistics** | |
| | **Model** | **Tolerance** | **VIF** |
| 1 | implementation | 0.950 | 1.052 |
| 2 | transparency | 0.950 | 1.052 |

[a] Dependent variable: innovation.

## 3. Methods

Our data descriptor sampling method followed the probability-sampling approach by means of a simple random technique that enables the selection of individual participants within the population sampling frame. In a simple random sampling method, every company within our study population had an equal chance of inclusion in the sampling frame. We calculated the sampling size using the probability factor for population (P) formula, as follows:

$$P = 1 - (1 - (1/N))n$$

where N represents the main population, and n represents the sampling size. Applying this formula with values of P = 0.10, and N = 467, we obtained a value of n = 47. By such means, we randomly selected 47 companies from our population of 467 SMEs [7]. We issued our questionnaire to these 47 SMEs and we collected our data in August 2022. We received 112 responses from 23 companies. Two of these were excluded as their responses were irrelevant to the research inclusion criteria. All participants in this sample were asked to describe their experience within the medical devices sector, as well as their own positions within the study SMEs, to ensure the quality of our survey.

We measured mentoring in such areas as the development of new MDR implementations. We also measured the overall output of SMEs, and the consequences of applying the contemporary restrictions in terms of the organization, supervision, and development of company personnel, as well as effects upon scientific research. Each broad category into which the questionnaire was organized was composed of several relevant questions [3,4]. For our data descriptor, we applied exploratory factor analysis (EFA) to assess the evidence of construct validity.

After EFA, and descriptive statistics analysis, we computed collinearity test results to investigate the validity of the data set, and to enable correlation with the inferential statistics of future research studies.

## 4. User Notes

The data descriptor conducted validity and descriptive analysis tests for the collected data set. The study statistically identified the negative effects of new regulatory protocols on the business performances of SMEs involved with medical devices in southern Germany, over and above the impacts of the contemporary economic situation. These data findings represent solid evidence of the negative effects of the MDR regulations within the medical devices sector. More positively, these data might be used by the regulatory affairs departments and strategic managerial sections of SMEs to develop a process to tackle the challenges caused by MDR. Moreover, the data can be used by management information systems (MIS) and decision system support professionals for planning and developing practical models about how to cope with current industry challenges Our study identifies specific areas of regulation that might be reconsidered and improved by regulators. In addition, researchers in the field of health economics might wish to conduct inferential analysis and triangulation of these data with a semi-structured qualitative study because we computed a multicollinearity test for the independent variables. We found variances inflation factor (VIF) values of 1.05 for both our independent variables, which means these variables are suitable for computing predictable data analysis, and thus of potential value in future research. Researchers in health economics might utilize these data to construct business models to predict the future performance of SMEs. They might also be used to analyze the relationship between poorly implemented regulations and poor organizational performance. In conclusion. based on our analysis, we make two recommendations as follows: first, the negative effects of MDR should be addressed by reconsidering the implementation process and by increasing the transformation process and deadline timeframes. Second, the classification of medical devices should be redefined, with a new priority list reflecting risk factors for each category, and an ascending process for obtaining the certificate of conformity (COC) with varying degrees of restriction and other requirements.

**Supplementary Materials:** The following supporting information can be downloaded at: https://www.mdpi.com/article/10.3390/data7110158/s1, MD SPSS Correlation, MDR Factor Tables, MED collinearity, MED extraction after removing factors, R results script, Survey about for medical devices (SMEs) in Germany (1–112).

**Author Contributions:** Conceptualization, H.A.E. and S.K.; methodology, H.A.E.; software, H.A.E.; validation, H.A.E., S.K. and N.E.-B.; formal analysis, H.A.E. and N.E.-B.; investigation, S.K.; resources, H.A.E., N.E.-B. and S.K.; data curation, H.A.E.; writing—original draft preparation, H.A.E.; writing—review and editing, H.A.E.; visualization, N.E.-B.; supervision, H.A.E.; project administration, H.A.E., N.E.-B. and S.K. All authors have read and agreed to the published version of the manuscript.

**Funding:** This research received no external funding.

**Informed Consent Statement:** The authors stated all ethical considerations and acknowledgements in the survey introductions, and all the respondents were fully aware that the purpose of this study was for academic publication only. We maintained full consideration of participant confidentiality.

**Data Availability Statement:** Submitted within supplementary files.

**Conflicts of Interest:** The authors declare no conflict of interest.

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
