# Peer review of "Measuring and Validating the Factors Influenced the SME Business Growth in Germany—Descriptive Analysis and Construct Validation"

_data, 2022_

Round 1

Reviewer 1 Report

This is an interesting article analyzing a few key factors which influence the business of SME Medical Device companies in Germany. The strength of this article is that this comes up with ample amount of quantitative analysis based on the structured survey. While this article is equipped with the strong quantitative analysis, the structure of this study needs to be strengthened to further elaborate the objective/background of the study. The following are specific comments for further improvement.

1. This article lacks the background/objective of the study. For readers, it is challenging to understand why the researchers attempted to tackle this topic. The article could briefly summarize the current situation around the medical device industry in Germany.

 2. This article could explain the academic contribution of the paper. By going through the achievements of existing papers, this article could mention that the study can make additional contribution to the relevant topics.

 3. Theoretical background can be another area which could be nicely added to this article. Though this article has mentioned a few statistical tools for analysis, several theories could be mentioned to further strengthen the fundamental theoretic background of the study.

 4. This article comes up with the three major variables which are business performance, financial situation, and medical regulation. However, this article lacks an explanation about why the researchers decided to choose these major variables. To do so, this article needs to have a relevant theory which could support the choice of the variables. In addition, the specific components of each variable need to be elaborated in the section of methodologies.

 5. MDR was highlighted as a specific variable for the field of medical regulation. Considering that not all the readers are knowledgeable about the regulation of medical devices, it could be better if this article provides readers with a proper amount of definition/explanation about the MDR regulation in CE approval process.

  6. Based on the result of the survey, this article stressed the negative impact of MDR regulation a few times. Instead of emphasizing the negative impact of MDR, the study could also discuss the areas of improvement of MDR to position this article as a source of constructive feedback for regulating bodies.

 7. The sections of both discussion and conclusion could be added if possible.

Reviewer 2 Report

This article brought data as resources with high value that should be validated with relevant and qualified processes. However, this article should be followed up by considering these items:

1. The paper structure should adapt the common format, such as IMRaD (Introduction, Method, Result, and Discussion).

2. Problem as initial point was not clearly narrated/described. Verify again the expectation and reality with adequate evidence, then narrate them using coherent premises.

3. Follow the standardization for the composition in Reference list.

Thank you

Round 2

Reviewer 1 Report

It is impressive that the author has quickly revised the article based on the feedbacks. While the article has achieved noticeable improvement, it is true that the article has limitation in effectively delivering its thoughts/implications mostly due to the lack of proficiency in the language. As we could understand that the limitation could come from the fact that the authors are not the native speaker of English, it will be very much desirable if the article could be intensively edited by an experienced proofreader. Once the article is edited nicely, it will have more impactful message delivery.

Author Response

Dear Valuable Editor,

Thanks for your guidance and recommendations to improve the quality of research paper. As per your comment the manuscript passed the proofreading and editing process attached you can find the certificate of editing by MDPI. once again thanks for your precious time to contribute the body of knowledge.

Reviewer 2 Report

Thank you for your commitment to response the feedback.

1. It's Ok if you followed the specific format for Data Descriptor. However, please complete the analysis, include the theoretical and practical implications.

2. Explain the methodology for data processing. As example you obtained the collected data through questionnaire based on likert-scale, then you processed them using PLS-SEM approach. Please state and explain basic step by step of the methodology/approach you used.

3. Check all references. Use the writing standard for them.

4.. In the end manuscript, state whether your data processing had solved the problems occurred or not.

Thank you so much

Author Response

"Please see the attachment

Round 3

Reviewer 2 Report

This progress is enough to decide tis acceptance. Overall, its quality was adequate. However, change "we" to "the authors" following basic style in the manuscript. Every formula also should be completed with citation or information about sources.

Thank you